# Multi-Risk Assessment to Evaluate the Environmental Impact of Outdoor Pig Production Areas: A Case Study

Carmo Horta [1,2,*] , Natália Roque [1,2,3,*] , Marta Batista [1] and António Canatário Duarte [1,2,4]

1   Polytechnic Institute of Castelo Branco, School of Agriculture, Quinta da Sra. De Mércules,
    6001-909 Castelo Branco, Portugal
2   CERNAS-IPCB Research Centre for Natural Resources, Environment and Society, Polytechnic Institute of
    Castelo Branco, 6001-909 Castelo Branco, Portugal
3   Qrural-Quality of Life in the Rural World-Research Unit, Polytechnic Institute of Castelo Branco,
    6001-909 Castelo Branco, Portugal
4   GEOBIOTEC Research Centre GeoBioSciences, GeoTechnologies and GeoEngineering, University of Beira
    Interior, Rua Marquês de Ávila e Bolama, 6201-001 Covilhã, Portugal
*   Correspondence: carmoh@ipcb.pt (C.H.); nroque@ipcb.pt (N.R.); Tel.: +351-272-339-900 (C.H. & N.R.);
    Fax: +351-272-339-901 (C.H. & N.R.)

**Abstract:** Outdoor pig production (OPP) can be considered an intensive system in many areas of the Mediterranean region. The concentration of the rainfall in the winter season, the OPP's topographic and soil properties, together with the continuous input of food and pigs' excreta, contribute to a profound increase in the nutrients leaching and soil erosion. This work aimed to evaluate the accuracy of the DRASTIC-LU index and the Revised Universal Soil Loss Equation (RUSLE) to provide early information to improve the planning of this type of pig production through more adequate location and sustainable management practices. The two models were applied to an OPP with 2.24 ha, with a heavy animal charge (one adult per 1.120 m$^2$). The results showed that 85% of the OPP area has a moderate risk to the vulnerability index to groundwater pollution and 15% high risk. The risk of soil erosion ranged from very severe to extremely severe in 96% of the area. The DRASTIC-LU indexes and the RUSLE model produce a multi-risk assessment that agreed with the observed field data. These two models showed accuracy to be used for early assessment when choosing the best location and improving management practices for OPP systems.

**Keywords:** DRASTIC-LU index; eutrophication; groundwater pollution; leaching; soil erosion; RUSLE equation

## 1. Introduction

The soil management used for outdoor livestock production should consider the benefits for consumers and the impacts on the environment of such use. The Mediterranean region offers good weather conditions for outdoor pig production (OPP), which is an animal production method with low capital costs to establish [1–3] and is considered more environmentally and animal-friendly than intensive indoor production. These are the main reasons for the increasing interest in this pig production system [4]. The effects of OPP in increasing soil organic matter (SOM) and nutrients are well reported [5–9]. In these outdoor production areas, cattle are fed with commercial feed, usually in fixed location feeders. These feeders incorporate high amounts of organic matter leading to adding nutrients into the soil. Cattle excretions are a source of organic matter and these nutrients are added to the soil. These continuous additions have an environmental impact on soils, leaching into groundwaters and causing losses through soil erosion, and they are usually substantial, turning the OPP into intensive livestock production. However, the environmental impacts depend on a set of factors. The soil properties, the slope of the area, the climate characteristics, and the groundwater depth are some of these factors that should be considered.

Moreover, according to Terranova et al. [10], in the Mediterranean environment, soils are particularly exposed to erosion by water for different reasons, such as inappropriate agricultural practices, deforestation, overgrazing, forest fires, and others. Under these climatic conditions, the soil is exposed to extended dry periods followed by heavy-erosion rainfall events on, many times, steep slopes and fragile soils. So, the soil-erosion process over time involves a lot of agro-environmental damage: (i) loss of fertile topsoil and, consequently, decrease of land productivity; (ii) on-site degradation of environmental quality and also beyond the location where it occurs; and (iii) decrease of soil organic matter and the good soil-related properties [11]. The simulation models play an essential function since they allow for improved soil management through adequate manipulation of the most sensitive parameters after being calibrated and validated for specific conditions [12]. Thus, the use of risk-assessment models, including these parameters, can provide data to help in decision-making to define the areas to establish this type of livestock production and use more environmentally-friendly strategies for the soil management of outdoor livestock systems.

One of the most widely used methods for assessing the intrinsic vulnerability of groundwater pollution is the DRASTIC model [13], as it is easy to compute with minimum data requirements. This model is based on the concept of the hydrogeological setting, defined as a composite description of the significant geologic and hydrologic factors affecting and controlling groundwater movement into, through, and out of the study area. The acronym represents seven hydrogeological parameters considered in the evaluation procedure. Thus, D represents the depth to groundwater, R the aquifer recharge, A the aquifer media, S the soil media, T the topography, I the impact of the vadose zone, and C the hydraulic conductivity of the aquifer. The addition of another parameter concerning land use/land cover is now also considered in the modified DRASTIC model (DRASTIC-LU) [14]. This DRASTIC model has been used in different studies to assess the aquifer vulnerability for regional development planning, to assess not only the impacts in intensive cultivation regions [15] but also the impacts of urban growth, agricultural activities [16], and even a country-scale assessment [17]. The soil erosion probability prediction by the Revised Universal Soil Loss Equation methodology [18] is another valuable model to improve management practices in OPP areas. Mapping soil erosion by the GIS-based RUSLE methodology has made it possible to assess the impact of different land uses on soil erosion and help to design more appropriate soil-management practices [19–21]. However, in the revised literature, we cannot find data about using these two models in areas with this livestock production system.

The behaviour of the pigs, continuously foraging the soil and keeping it bare through the year, increases the potential for soil erosion and leaching in principle more deeply than in other cultivated systems reported in the literature. In addition, the rainy and cold winters in the Mediterranean region favour the loss of nutrients and organic matter by leaching and erosion. So, in this work, we tested the hypothesis that to improve sustainable management practices for the OPP areas, the use of the DRASTIC-LU model together with the Revised Universal Soil Loss Equation (RUSLE) will allow us to assess the global vulnerability of the area to the groundwater pollution and the soil erosion risk. To evaluate the accuracy of this approach, we will show a case study from an OPP located in a Mediterranean region. For this OPP, we will test the DRASTIC-LU and the RUSLE models. The impact of this OPP on the spatial distribution of the soil organic matter and soil labile P as soil indicators related to the eutrophication of the waterbodies will also be shown. The prediction of soil loss potential by the RUSLE model will be compared with the observed soil loss in this OPP.

## 2. Material and Methods

### 2.1. Characterisation of the Experimental Unit of Outdoor Pig Production

The outdoor pig production area is located at a farm belonging to the School of Agriculture of the Polytechnic Institute of Castelo Branco, Portugal (39.822054, −7.448026 Decimal Degrees, WGS 84 coordinates). The OPP consisted of 2.24 ha divided into six paddocks

(Figure 1a). This OPP evaluated the behaviour of two local breeds, Alentejana and Bísara. So, in paddock 1 (Pk1, Figure 1a), there were two males, one of each breed, paddock 2 had the pregnant/lactating Alentejana females, paddock 3 had pregnant/lactating Bísara females, and in paddock four, there were the Bísara females. In paddock five there were the Alentejana females. Paddock 6 had the weaned piglets. The experimental area had, on average, a charge of one animal adult per 1.120 m². The OPP had 18 adult females, nine of each breed, which had two parturitions per sow per year with an average of five piglets per parturition. Piglets were sold after 60 days of birth. Average food intake per animal and day was between 2.5 and 3.5 kg of commercial concentrates with a P concentration in dry matter of 0.4%, but for pregnant/lactating females, the concentrate had 0.7% of P. This area was neither cultivated nor fertilised, so nutrients and organic matter inputs were due only to the feed and pig excreta. The feed and water points were fixed (Figure 1a) near the paddock's door and located in lower elevation spots inside each paddock. The slope of the area is shown in Figure 1b.

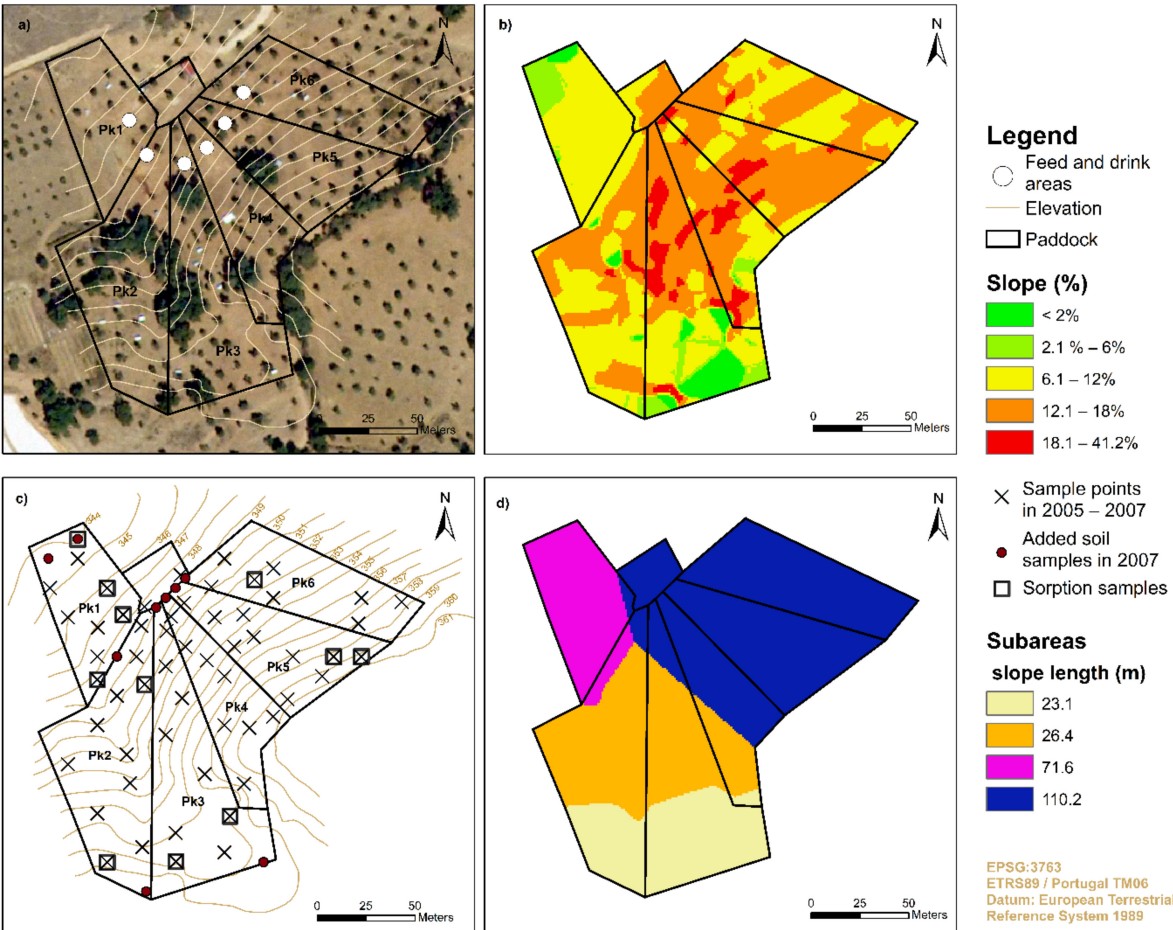

**Figure 1.** Outdoor pig production area, (**a**) paddocks, feeders and well points, (**b**) slope (%), (**c**) soil sample points and elevation of the area, and (**d**) homogenous areas with similar slope length (m).

The soil properties used to assess the environmental impacts of the OPP were the changes in the distribution of the content on soil OM and the labile soil P two years after the experiment's beginning (January 2005).

### 2.2. Characterisation of Climate, Data Acquisition, Land Use, and Soil Properties

The region has a Mediterranean climate, with an average (1986–2015) temperature of 15.0 °C, presenting a minimum value of 9.4 °C and a maximum value of 9.4 °C and 735 mm

of annual precipitation in which the month with the least precipitation is July with 5.9 mm and the month with the most precipitation is December with 112.2 mm [22].

The topographic data was collected in 2005 by classic topography, before the installation of the pig production and in 2022 by UAV (Drone) supported by 14 ground points whit millimetric precision. The first method produces isolines with a 1 m distance, and the second produces a DTM (Digital Terrain Model) with a pixel size of 0.02 m and is reclassified to the 1 m, so it was possible to obtain comparable data.

The land use at the beginning of the OPP consisted of open Agroforestry of *Quercus suber* L. and *Olea europea* L. with natural pasture. This ecosystem was at natural equilibrium.

Soil properties were evaluated by sampling soil as follows: 56 positioned randomised soil samples were taken in all the areas (2.24 ha) from a network of 5 m × 5 m at the beginning of the experiment. On 1 February 2007, another soil sampling was done in the same positioned soil samples, and another seven samples were collected in sites with visual variations of SOM (Figure 1c). This procedure allowed us to identify during the experiment the variation (increase/accumulation or decrease/loss) on SOM and soil labile P. Depth of soil sampling was always 0.20 m. The soil samples were taken in February 2007 and were analysed for organic matter (OM) and labile P quantified by the Olsen procedure (Olsen-P).

Background soil properties were evaluated before the beginning of the experiment in January 2005. Soil samples were air-dried and sieved (<2 mm). The pH was measured using a pH electrode by taking 10.0 g of dried and sieved soil and 25 mL of distilled water (1:2.5 soil: solution ratio) which were in contact for one h. According to the procedure described in Walkley and Black [23], the organic matter was analysed using a potentiometric titration method [24]. The soil texture was evaluated by the particle size analysis using the pipetting method [25]. The procedure of Olsen et al. [26] was used to quantify the labile soil P (Olsen-P), i.e., not only the soil P available to the crops but also the transfer from soil to runoff waters [27]. The content of P in all the extracts was quantified by the method of Murphy and Riley [28].

Variability between the initial 56 soil samples was very low, so we took the average values to characterise the background soil properties. The soil of the experimental area is a Dystric Cambisol [29], sandy loam (clay 11.4, silt 10.3, and sand 78.3%), acid (pH = 5.1), poor in organic matter (1.4%) and with a low level of labile P (5 mg kg$^{-1}$).

*2.3. DRASTIC—LU Model*

The DRASTIC model [13] was developed by the US Environmental Protection Agency (EPA) to evaluate the groundwater contamination potential for the entire United States. Fritch et al. (2000) first modified the model and incorporated a parameter to allocate the Land use/Land cover in the DRASTIC-LU model. This modification integrated the soil's impact on the soli's vulnerability, with each parameter being evaluated against the others to determine the relative importance of each and then assigning a relative weight, ranging from 1 to 5 (Table 1). The most significant parameters are given a weight of 5, while the least significant receive a weight of 1 [13]. The purpose index involves multiplying each factor's weight (Table 1) by its category rating (Table 2), where rating (r) reflects the significance of classes and weight (w) designates the importance of the parameter, as follows:

$$DRASTIC - LU = (D_r \times D_w) + (R_r \times R_w) + (A_r \times A_w) + (S_r \times S_w) + (T_r \times T_w) + (I_r \times I_w) + (C_r \times C_w) + (LU_r \times LU_w) \quad (1)$$

where $D$ is the depth to groundwater (m), $R$ is the recharge rate (net) (mm), $A$ is the aquifer media, $S$ is the soil media, $T$ is the topography (slope) (%), $I$ is the impact of the vadose zone, $C$ is the conductivity (hydraulic) of the aquifer (m day$^{-1}$), and $LU$ is the Land use/Land cover (Equation (1)).

**Table 1.** Assigned weights for DRASTIC-LU parameters.

| Parameters | Weight |
|---|---|
| Depth | 5 |
| Recharge | 4 |
| Aquifer media | 3 |
| Soil media | 2 |
| Topography | 1 |
| Impact of vadose zone | 5 |
| Hydraulic conductivity | 3 |
| Land Use/Land Cover | 3 |

**Table 2.** DRASTIC-LU parameters.

| Parameter | Range | Rating |
|---|---|---|
| Depth to groundwater | 1.63–4.6 m | 9 |
| | 4.6–9.1 m | 7 |
| Recharge rate (net) | >300 (mm) | 8 |
| Aquifer media | Metamorphic rocks | 3 |
| Soil media | Medium | 9 |
| | <2% | 10 |
| | 2–6% | 9 |
| Topography (slope) | 6–12% | 5 |
| | 12–18% | 3 |
| | >18% | 1 |
| Impact of the vadose zone | Metamorphic rocks | 4 |
| Hydraulic Conductivity of the aquifer | <4.1 (m day$^{-1}$) | 1 |
| Land Use/Land Cover | Barren Land | 8 |

The data, depth to groundwater, the topography, and the hydraulic conductivity of the aquifer was acquired at the local scale, and data for recharge rate, soil, and land use were incorporated at the lowest scale available to improve the model accuracy.

### 2.4. Assessment of Soil Erosion by the Revised Universal Soil Loss Equation

The RUSLE equation [19], and its predecessor USLE (Universal Soil Loss Equation) [29], calculate the average annual soil loss by the following multiplicative expression:

$$A = R \times K \times L \times S \times C \times P \tag{2}$$

where the factors assumed the following values for this case:

*A* is the annual soil loss (ton ha$^{-1}$ year$^{-1}$), *R* is the precipitation and runoff erosivity factor (1173 MJ mm ha$^{-1}$ h$^{-1}$ year$^{-1}$), *K* is the soil erodibility factor (0.042 t h MJ$^{-1}$ mm$^{-1}$), and *L* is the factor for slope length (dimensionless; variable within the area of study). *S* is the slope steepness factor (dimensionless; variable within the area of study), *C* is the factor for land cover and cropping practices (0.5; dimensionless), and *P* is the soil conservation practices (1.0; dimensionless). The value of the rainfall erosivity (1173 MJ mm ha$^{-1}$ h$^{-1}$ year$^{-1}$) is an average value based on 14 years of rainfall data taken in a meteorological station located very close (1 km) to the location of the paddock, and according to the EI30 Index methodology [29]. Since the data acquisition was obtained from a local Meteorological station, the precipitation, the runoff erosivity factor, and the soil erodibility factor were local measurements that increased the model's accuracy. Also, the topographic models from 2005 and 2022 were obtained with digital data, and both were standardised at 1 m × 1 m raster.

The application of the RUSLE model to this study area was based on the following simplifications since it is a small area with the same soil use (intensive pig grazing) for the continuous analysis period. So, the *R*, *K*, *C*, and *P* factors assume the same value, and for the topographic factor (*LS*), the total area is divided into four subareas with similar slope

lengths (Figure 1d), and the land slope is calculated by ArcGis 10.8 tool inside of each cell by the RUSLE [19].

*2.5. Data Analysis*

The changes (Δ, %) in SOM and soil labile P from January 2005 to February 2007 were evaluated by the following equation:

$$\Delta\ (\%) = 100 \times \frac{(\text{soil value Feb. 2007} - \text{background soil value})}{\text{background soil value}} \qquad (3)$$

The maps with the spatial distribution of these changes (Δ, %; Equation (3)) of SOM and Olsen-P were done using the georeferenced data of the soil samples with ArcGIS 10.8.

To process the datasets and compute the weight input variables in the DARASTIC-LU, ArcGIS 10.8 was also used. Depth point information was collected from the Spanish and Portuguese piezometers network using the dataset from 1986 to 2020 in 23 locations [30,31]. The Inverse Distance Weighted (IDW) algorithm was used for interpolation purposes, and, finally, a raster data file was produced corresponding to a continuous Depth layer along the surveyed area. The Recharge rate layer was obtained from [32], expressed in mm year$^{-1}$. Aquifer media and the Impact of the vadose zone are influenced by the Metamorphic rock's geological information [33]. Soil media was inferred from [34]. Topography data is driven by the topographical study (isolines distanced 1m). The Hydraulic conductivity (Ks) was calculated according to [35,36] and is explained in Supplementary Material S1, and the value obtained was 2.24 m day$^{-1}$. Land Use was incorporated following the data and ranking referred to by [37]. The obtained layers were reclassified and transformed into raster files, considering the indicated ranges and rates (Table 2). The vulnerability index obtained from the DRASTIC-LU model classifies the areas matching the vulnerability classes defined by [13].

**3. Results**

*3.1. Drastic-LU Model*

The DRASTIC-LU *vulnerability index* [13] was constructed using Equation (1). The obtained results were reclassified in different classes, corresponding to the *vulnerability index* classified as Moderated (120–149) in 85% of the OPP and high vulnerability index (150–179) in 15% of the area (Figure 2 and Table 3).

*3.2. Evaluation of Soil Losses by Water Erosion*

3.2.1. Soil Erosion Prediction by the RUSLE Equation

The simulation of soil erosion by the RUSLE model shows that the majority of the study area is included in the class [38] very severe (16.5%) and extremely severe (79.7%) (Table 4 and Figure 3a) together. These percentages represent 96.2% of the total area. The amount of soil loss average in the whole area is 502.4 t ha$^{-1}$ year$^{-1}$, representing an extremely high value and an average loss of effective soil depth of 0.56 m in the period of analysis (17 years). As was highlighting [39] by Schnabel et al. (2009) about the negative effect of excessive livestock density, this value was expected, considering the form of pigs grazing and the area to be very small for the number of pigs (20 adults along the year, and an average of 90 piglets during two months for each parturition). In extensive regimen, often under *Quercus* forest, this value is much lower and often inside the concept of sustainability [40].

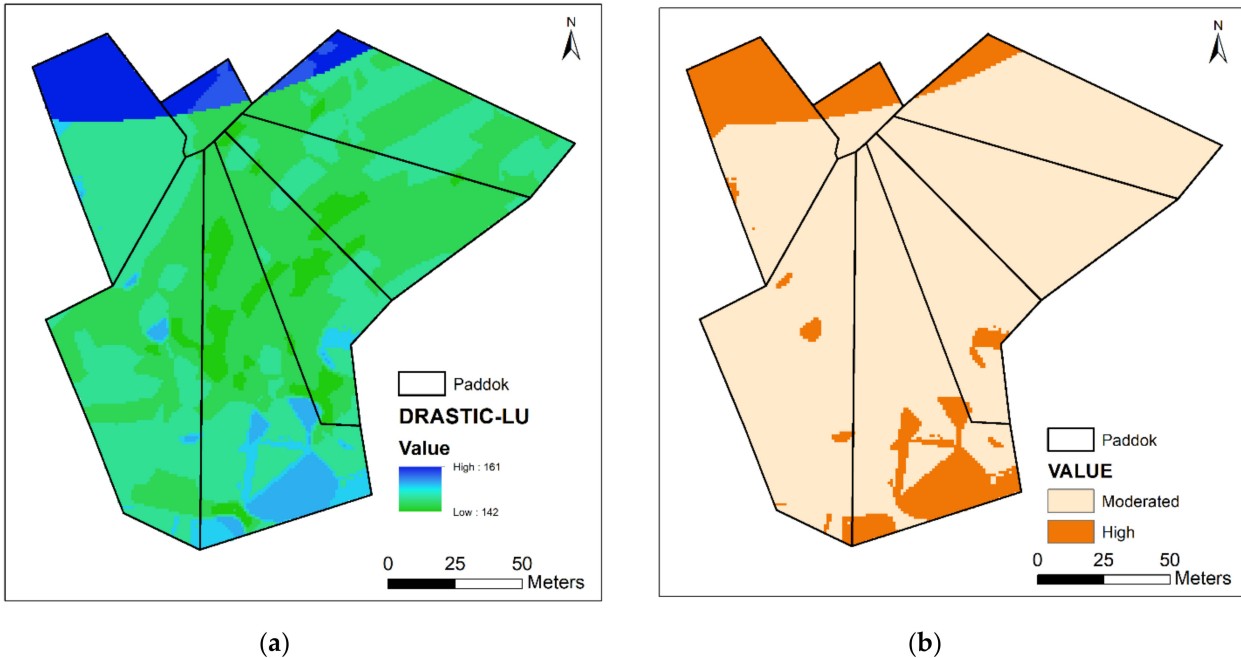

(**a**)                                                                                        (**b**)

**Figure 2.** Spatial distribution of the *vulnerability index* obtained by the DRASTIC-LU model, (**a**) continuous variation, and (**b**) classes of the *vulnerability index*.

**Table 3.** The DRASTIC-LU model in the OPP obtained the vulnerability index.

| DRASTIC | Area (%) | Vulnerability Index |
|---------|----------|---------------------|
| 120–149 | 85 | Moderated |
| 150–179 | 15 | High |

**Table 4.** Soil loss prediction obtained by the RUSLE model observed soil loss and erosion risk classes adapted from [38] in the OPP.

| Soil Erosion (t ha$^{-1}$ y$^{-1}$) | RUSLE Area (%) | Observed in OPP Area (%) | Erosion Risk Classes |
|---|---|---|---|
| <5 | 0.2 | 0.1 | Very low |
| 5–12 | 0.2 | 0 | Low |
| 12–50 | 0.6 | 0.5 | Moderate |
| 50–100 | 2.8 | 0.6 | Severe |
| 100–200 | 16.5 | 1.7 | Very severe |
| >200 | 79.7 | 93.7 | Extremely severe |

3.2.2. Soil Erosion Observed in the Field

The observed soil loss in the study area was evaluated by comparing the difference between the altitudes of two Digital Elevation Models (DEMs), one relative to the topography from 17 years ago and the other relative to the topography in the present (February 2022). As we can see in Figure 4, the majority of the area (68.5%) presents a loss of effective depth between 0.5 and 0.99 m, but there are areas with extreme loss of effective depth (1.99–2.25 m; 0.2%) and others with soil sedimentation (0.0–1.2 m; 2.0%).

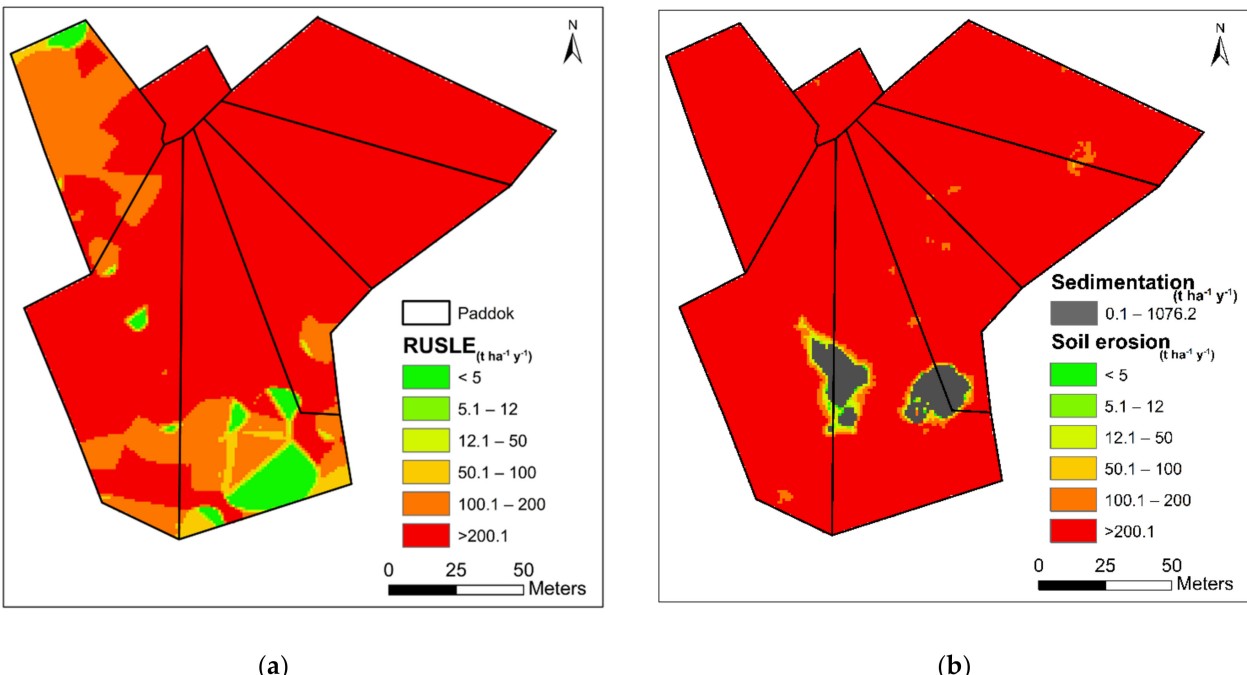

(**a**)                                                                                  (**b**)

**Figure 3.** Soil erosion (t ha$^{-1}$y$^{-1}$), (**a**) prediction by the RUSLE model and (**b**) observed soil erosion and sedimentation.

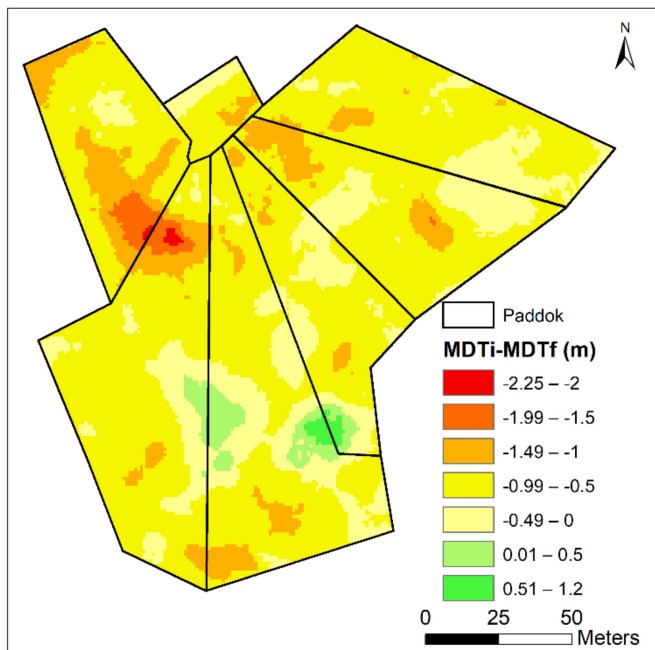

**Figure 4.** Location in the study area of sites with soil loss (m; negatives values) and sites with sedimentation (m; positive values).

The first situation refers to areas where the food and drink sites are located and, by observation, sites where the animals stay for more extended periods and, consequently, more pitted and excavated. The second situation concerns two concave zones flanked by very sloped hills, generating high rates of erosion and sedimentation in the concave zones.

Concerning the observed soil loss (Figure 3b), considering some previous comments, most of the area showed an extremely severe erosion rate (93.7%) and very severe (1.7%). The remaining areas (4.6%) with a nonexistent or moderate erosion rate coincide with the areas where sedimentation occurs or in the borders. These zones are located near the head

of the streams, with a small runoff volume, Table 5. The sedimented soil remains in these sites. The average observed soil loss in the whole area is 609.7 t ha$^{-1}$ year$^{-1}$, representing an extremely high value and an average loss of effective soil depth of 0.68 m in the period of analysis (17 years).

**Table 5.** Soil sedimentation observed in the OPP.

| Soil Sedimentation (t ha$^{-1}$ y$^{-1}$) | Observed in OPP Area (%) |
| --- | --- |
| <5 | 0.1 |
| 5–12 | 0.4 |
| 12–50 | 0.5 |
| 50–100 | 0.5 |
| 100–200 | 0.7 |
| >200 | 1.2 |

*3.3. Impact of the OPP on SOM and Soil Labile P*

The positioned soil samples allowed the evaluation of the changes on SOM and on labile soil P (Olsen-P) in the short time (two years after the beginning of the OPP) (Figure 5). An overall SOM accumulation was observed above the background value (SOM = 1.4%), with SOM values ranging from 2.6 to 6.6% and an increase in the background soil value ranging between 85 to 376%.

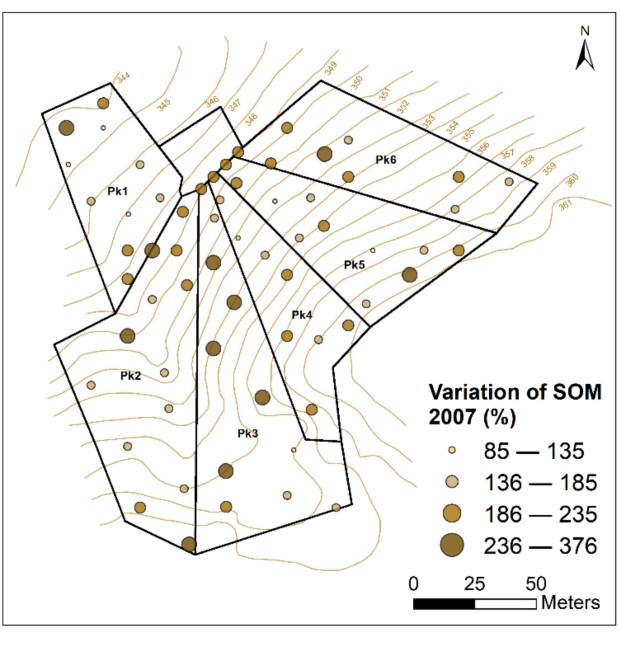

(**a**)

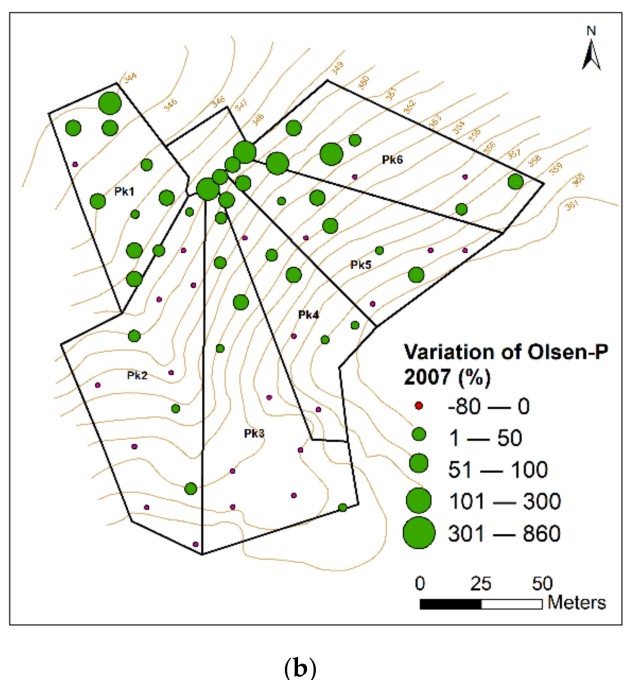

(**b**)

**Figure 5.** Spatial distribution of the variation (%) of (**a**) SOM and (**b**) Olsen-P (labile P) after two years from the beginning of the OPP.

The soil Olsen-P showed spots with P losses and accumulation in relation to the background value (Olsen-P = 5 mg kg$^{-1}$). Thus, in the same period of time, the Olsen-P ranged from 1 to 49 mg kg$^{-1}$ with a variation between $-82$ to 884%.

According to Fernández et al. [41], pigs can excrete 1.2 kg P for each 100 kg of live weight, with the growing pigs contributing 59% to P excretion, sows with 26%, and weans with 15%. Considering the live weight of the pigs in our OPP (data not shown) in paddock 6, for example, the total P input m$^{-2}$ in this period was around 190 kg P.

## 4. Discussion

The OPP showed an overall moderate vulnerability index to groundwater pollution, with some spots with a high vulnerability index. The high values of the DRASTIC-LU index are located in the OPP areas, with the most significant soil accumulation observed in labile phosphorus, organic matter, and sediment loss through erosion.

Concerning the results of the soil loss by water erosion observed and simulated by the RUSLE model, it is worth noting that the distribution of the erosion rate in the study area is similar and was primarily dominated by the rate extremely severe. This livestock production system proves to have an excessive animal charge ha$^{-1}$. As was noted by some authors, overgrazing, or unsustainable grazing, is one of the most acute problems in land management on a global scale and is regarded as a serious pressure on natural landscapes [42–44]. By comparing the observed and simulated erosion rate, we can infer that the RUSLE model can be used to predict average soil loss in intensive pig grazing in the total study area and over a long period of time. However, the model did not demonstrate the same capacity to simulate the erosion process in sub-areas inside the whole area. It may be due to the exactitude of the DEM referent to 17 years ago, which was based on rasterisation of the topographic maps, and, therefore, less accurate [45]. Another reason can be the inability of the RUSLE model to simulate the sedimentation that occurs in the study area, which is common [18].

Regarding the spatial distribution of SOM and labile P through the OPP, an overall accumulation can be observed above the background soil values (Figure 5). However, the higher SOM and P accumulation values occurred mainly at lower elevation spots corresponding to the surrounding area of feeders and wells. This feature can be explained by the preferential pathways for sediments and runoff water transport in the area, as observed in Figures 3 and 5. In addition to the slope and topographic characteristics, the daily feed inputs and the pig's behaviour also strongly influenced the spatial heterogeneity of SOM and soil P. For example, Eriksen and Kristensen [46] also found a high correlation between soil N, P and K accumulation and the distance to feeders. Watson et al. [47] refer that the preferred areas for excretion are out of the feeding zone and near the boundary of the paddocks. They observed in those select areas a P concentration in the top 0.05 m, on average six times greater than in the least preferred areas, and a large proportion of the increase in soil P seems to be associated with organic P forms. Salomon et al. [48] pointed out that 43 to 95% of nutrients were concentrated in preferred areas corresponding to 4–24% of the total pen area, with an increase of P on the topsoil of the preferred areas more than fourfold. The high amount of P input to soil together with the rainfall, which in this OPP occurs mainly from October to May, contributed to the downward movement of SOM and labile P not only in particulate forms (by erosion) but also in dissolved forms (in drainage or runoff waters). It was also observed that the content of dissolved P in drainage waters evaluated at 0.60 m depth was, on average, 0.1 mg L$^{-1}$ with maximum values of 0.3 mg L$^{-1}$ (data not shown). The threshold value of groundwater quality is 0.1 to 0.2 mg L$^{-1}$ [49,50]. So, the drainage waters in this area are a source of P to groundwaters increasing the risk of eutrophication of water bodies. Labile P showed high mobility in the paddocks as showed by its broad changes in soil Olsen-P content (−82 to 884%)

Although we cannot find values of P losses in runoff waters from OPP areas in the published research works, Watson et al. [47] referred that after 15 months of outdoor pig production, the soil profile was saturated with P and represented a significant environmental risk. Also, Sharifi et al. [7] observed the movement down to soil profile of Olsen-P, organic and total P at an outdoor area of farrowing sows. In addition, the work of Horta and Torrent [26] regarding the acidic Portuguese soils indicated an Olsen-P level of 20 mg kg$^{-1}$ or 50 mg kg$^{-1}$ as the threshold level to prevent a significant increase of losses of P from soils to drainage waters or runoff. So, this OPP showed that even in a short time, it has a high potential to contribute to the pollution of groundwater and losses of sediments by erosion. This observation agrees with the results obtained by the DRASTIC-LU and the RUSLE models used in the multi-risk assessment of this OPP.

## 5. Conclusions

The pollution risk of the groundwater provided by the Drastic-LU index showed that 85% of the OPP area has a moderate *vulnerability index* and 15% with a high *vulnerability index*. Around 96% of the OPP area has a risk of soil erosion from *very severe* to *extremely severe*, as estimated by the RUSLE model and from the observed data.

The observed results about the spatial distribution of the SOM and the labile P, and the observed soil erosion, agree with the results obtained by the RUSLE and the DRASTIC-LU models used in the multi-risk assessment of this OPP area.

From the results of the DRASTIC-LU *vulnerability index*, the soil erosion risk and the soil accumulation of OM and labile P, it is evident the unsustainability of this pig livestock system due to the evident soil damage: (a) in-site, related to the extremely severe erosion rates and very high loss of effective soil depth; (b) off-site, regarding the damage of downstream in the surrounding waterbodies, mainly by silting of the stream's bed and some hydraulic structures, and to the risk for groundwater pollution and eutrophication of superficial waterbodies. The DRASTIC-LU *vulnerability index* and the RUSLE models showed accuracy to be used as *vulnerability indexes* to define the areas to establish livestock outdoor production and planning actions to improve less degradative practices for the soil and water resources.

**Supplementary Materials:** The following supporting information can be downloaded at: https://www.mdpi.com/article/10.3390/agronomy12081898/s1, Supplementary Material S1 with the calculation of *Ks* [35,36,51–54].

**Author Contributions:** Conceptualisation, C.H.; methodology, C.H., N.R. and A.C.D.; performed the soil experiments, M.B.; analysed the data, C.H., N.R., M.B. and A.C.D.; writing, C.H., N.R. and A.C.D. All authors have read and agreed to the published version of the manuscript.

**Funding:** This research was funded by National Funds through FCT—Foundation for Science and Technology under the Projects UIDB/00681/2020 and CERNAS-IPCB.

**Institutional Review Board Statement:** Not applicable.

**Informed Consent Statement:** Not applicable.

**Data Availability Statement:** The data supporting this study's findings are available upon reasonable request.

**Acknowledgments:** The authors acknowledge the support of the School of Agriculture of the Polytechnic Institute of Castelo Branco and the Interdisciplinary Centre of Languages, Cultures and Education, of the Polytechnic Institute of Castelo Branco.

**Conflicts of Interest:** The authors reported no potential conflict of interest.

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
