# Peer review of "Multi-Risk Assessment to Evaluate the Environmental Impact of Outdoor Pig Production Areas: A Case Study"

_agronomy, doi:10.3390/agronomy12081898_

Round 1

Reviewer 1 Report

In its current form, critical improvements to your research methodology are required to ensure your results are appropriately substantiated and conclusions are fully supported, in alignment with current research standards. The current manuscript has a very local focus, not suitable to a wider international audience.

Author Response

Review Report Form

Revision

Does the introduction provide sufficient background and include all relevant references? Must be improved

Without comments duly supported on the Article under review the authors cannot reply.

Are all the cited references relevant to the research? Must be improved

Without comments duly supported on the Article under review the authors cannot reply.

Is the research design appropriate? Must be improved

Without comments duly supported on the Article under review the authors cannot reply.

Are the methods adequately described? Must be improved

Without comments duly supported on the Article under review the authors cannot reply.

Are the results clearly presented? Must be improved

Without comments duly supported on the Article under review the authors cannot reply.

Are the conclusions supported by the results? Must be improved

Without comments duly supported on the Article under review the authors cannot reply.

 Comments and Suggestions for Authors

In its current form, critical improvements to your research methodology are required to ensure your results are appropriately substantiated and conclusions are fully supported, in alignment with current research standards.

We have applied current research standards, for example in the key factor because the topographic support data refer to the current year and provide us with detailed data (20 cm resolution), of very high precision, it is these data that provide the accuracy in the two models.

The current manuscript has a very local focus, not suitable to a wider international audience.

Our methodology is applicable in wider international audience, because this reproduceable and replicable, mainly because the producers are engaged in more sustainable and environmentally friendly forms of production, and it is mandatory for the scientific community to develop tools and data that can support the decision where these productions can and will be less harmful to the environment.

Reviewer 2 Report

when the design of the MS is not correct, the entire MS will be not correct

Author Response

Review Report Form

Revision

Moderate English changes required

The English was revised by a native.

Does the introduction provide sufficient background and include all relevant references? Must be improved

Without comments duly supported on the Article under review the authors cannot reply.

Are all the cited references relevant to the research? Must be improved

Without comments duly supported on the Article under review the authors cannot reply.

Is the research design appropriate? Must be improved

Without comments duly supported on the Article under review the authors cannot reply.

Are the methods adequately described? Must be improved

Without comments duly supported on the Article under review the authors cannot reply.

Are the results clearly presented? Must be improved

Without comments duly supported on the Article under review the authors cannot reply.

Are the conclusions supported by the results? Must be improved

Without comments duly supported on the Article under review the authors cannot reply.

Comments and Suggestions for Authors

when the design of the MS is not correct, the entire MS will be not correct

We have applied current research standards, for example in the key factor because the topographic support data refer to the current year and provide us with detailed data (20 cm resolution), of very high precision, it is these data that provide the accuracy in the two models.

Reviewer 3 Report

TITLE: MULTI-RISK ASSESSMENT TO EVALUATE THE ENVITONMENTAL IMPACT OF OUTDOOR PIG PRODUCTION AREAS: A CASE STUDY

GENERAL COMMENT

The topic of this manuscript is interesting, and it is well-written and well organised. In my honest opinion, authors have performed good research and they have reflected it in the written manuscript in a concise way.

SPECIFIC COMMENTS

[line 126] Please, you must also specify the minimums and maximums of temperature and annual rainfall.

Please, you should revise the English style/grammar of the whole paper.

OVERALL MERIT

Accept after minor changes.

Author Response

The reviewer

Revision

SPECIFIC COMMENTS

 [line 126] Please, you must also specify the minimums and maximums of temperature and annual rainfall.

The region has a Mediterranean climate, with an average (1986-2015) temperature of 15.0ºC, presenting a minimum value of 9.4 ºC and a maximum value of 9.4 ºC and 735 mm of annual precipitation in which the month with the least precipitation is July with 5.9 mm and the month with the most precipitation is December with 112.2 mm [22].

Revise the English style/grammar of the whole paper.

The English was revised in spatial the grammar stile by a native.

Round 2

Reviewer 2 Report

Again, I did not see any change!!

In my opinion to accept this MS to publish, the authors need to repeat this experiment as a long-term study

Then they can compare their results in 2006 and what they can get in 2022

They can in this case publish a very interesting study using the old results as well

I am sorry to reject it again